# Activated Carbon from Palm Date Seeds for CO_2_ Capture

**DOI:** 10.3390/ijerph182212142

**Published:** 2021-11-19

**Authors:** Amira Alazmi, Sabina A. Nicolae, Pierpaolo Modugno, Bashir E. Hasanov, Maria M. Titirici, Pedro M. F. J. Costa

**Affiliations:** 1Department of Chemistry, University Colleges at Nairiyah, University of Hafr Albatin (UHB), Nairiyah 31981, Saudi Arabia; 2School of Engineering and Materials Science, Queen Mary University of London, London E1 4NS, UK; sabina.nicolae@qmul.ac.uk (S.A.N.); p.modugno@qmul.ac.uk (P.M.); 3Physical Science and Engineering Division, King Abdullah University of Science and Technology (KAUST), Thuwal 23955-6900, Saudi Arabia; bashir.hasanov@kaust.edu.sa (B.E.H.); pedro.dacosta@kaust.edu.sa (P.M.F.J.C.); 4Department of Chemical Engineering, Imperial College London, London SW7 2AZ, UK; m.m.titirici@qmul.ac.uk

**Keywords:** hydrothermal carbonization, activation, adsorption, palm date seeds, CO_2_ capture

## Abstract

The process of carbon dioxide capture and storage is seen as a critical strategy to mitigate the so-called greenhouse effect and the planetary climate changes associated with it. In this study, we investigated the CO_2_ adsorption capacity of various microporous carbon materials originating from palm date seeds (PDS) using green chemistry synthesis. The PDS was used as a precursor for the hydrochar and activated carbon (AC). Typically, by using the hydrothermal carbonization (HTC) process, we obtained a powder that was then subjected to an activation step using KOH, H_3_PO_4_ or CO_2_, thereby producing the activated HTC-PDS samples. Beyond their morphological and textural characteristics, we investigated the chemical composition and lattice ordering. Most PDS-derived powders have a high surface area (>1000 m^2^ g^−1^) and large micropore volume (>0.5 cm^3^ g^−1^). However, the defining characteristic for the maximal CO_2_ uptake (5.44 mmol g^−1^, by one of the alkaline activated samples) was the lattice restructuring that occurred. This work highlights the need to conduct structural and elemental analysis of carbon powders used as gas adsorbents and activated with chemicals that can produce graphite intercalation compounds.

## 1. Introduction

There is a pressing concern with the global warming and climate change occurring on our planet. For almost a century, the continuous combustion of fossil fuels has led to a measurable increase in the atmospheric concentration of carbon dioxide (CO_2_), a well-known greenhouse gas. In order to mitigate this issue, a number of strategies have been proposed, chief amongst which are CO_2_ capture technologies. Reversible gas adsorption (physisorption) is a mature field that relies on the appropriate selection of porous materials used as adsorbents to capture a particular gas [1,2,3,4,5]. In the case of CO_2_, a number of physical adsorbents have been investigated, such as activated carbon [6,7], mesoporous silica [8], zeolites [7,9], metal–organic frameworks [10] and fly ash [11]. Of these, high-surface-area powders of porous carbons present a number of notable advantages. In addition to the ease of synthesis and regeneration, they show remarkable chemical and thermal stability [12]. Sustainable porous carbon materials can be obtained from waste biomass through various carbonization and activation processes. A popular option is performing hydrothermal carbonization (HTC) followed by chemical activation with strong oxidisers [13,14].

In Saudi Arabia, palm trees are very abundant. With more than 23 million specimens, the aggregated production of palm date fruits in the Kingdom reached a staggering 1,078,300 metric tonnes in 2010, which is equivalent to 14.4% of the world’s production [15]. Regrettably, there is very little use for the seeds (or pits) of these fruits, which are commonly considered waste and mostly end up in landfills or incinerated. This poses a serious environmental challenge, as it is estimated that the palm date biomass waste amounts to almost one million metric tonnes per year in Saudi Arabia alone [16,17].

Recently, the utilisation of biomass waste to produce low-cost sorbents for CO_2_ capture has attracted significant attention [18]. With respect to palm date seeds (PDS), there is limited information on the application of this natural resource [19,20,21,22,23]. In fact, the natural structure, lignocellulose composition and low ash content of PDS makes it an excellent precursor for activated carbon [19]. In average, the mass of PDS is composed of cellulose (42%), hemicellulose (18%), sugar and other compounds (25%), lignin (11%) and ash (4%) [24].

In this work, we investigate the performance of activated carbons derived from palm date seeds for CO_2_ capture. To augment the PDS char porosity, endogenous palm date fruits (*Phoenix dactylifera*) were subjected to HTC followed by chemical activation. Three activation routes were explored: (1) gaseous oxidation, (2) alkaline wet impregnation and (3) acid wet impregnation.

## 2. Experiment

### 2.1. Preparation of the PDS Hydrochar

The pits of *Phoenix dactylifera*, also known as the date or palm date, were collected from farms near the city of Damman, Saudi Arabia. After cleaning, the palm date pits were pulverised and used as the biomass source for the production of porous carbon materials. In a typical synthesis, 5 g of the powdered pits was initially dispersed in 50 mL of deionised water. This mixture was transferred into a stainless-steel autoclave and heated at 200 °C for 48 h, resulting in the hydrothermal carbonization (HTC) of the palm date pit biomass. Upon cooling to room temperature, a dark-brown precipitate was separated by vacuum filtration, washed with deionised water (until the pH was neutralised) and dried at 100 °C overnight. Hereafter, this powder is denoted PDS hydrochar (or simply, HTC-PDS).

### 2.2. Activation of the PDS Hydrochar

Given the lack of porosity, the powdered PDS has low surface area making its hydrochar inappropriate for gas uptake studies [25,26,27,28]. Hence, post-carbonization steps were developed to augment the porosity. Three independent activation processes were studied. The first (1) was CO_2_ activation: typically, a certain mass of the PDS hydrochar was placed in a ceramic crucible, covered with the respective lid and placed in a tubular furnace under a CO_2_ flow rate of 50 cm^3^/min. The samples were heated to 900 °C at a rate of 5 °C/min and exposed to the oxidative gas stream for 3 h. The product was denoted HTC-PDS_CO_2_ activation. The second activation process (2) was an alkaline-medium chemical activation (with KOH): two different mass ratios of hydrochar:KOH were investigated, 1:1 and 1:2. Initially, a wet impregnation was performed by dispersing the PDS in a concentrated aqueous solution of KOH and stirring for about 2 h. After drying the KOH-impregnated PDS powder in an oven (overnight, 100 °C), a high-temperature treatment was carried out for 2 h (dwell time) under an inert atmosphere (N_2_, 99.995%, flow of 150 cm^3^/min) at 900 °C (with 5 °C/min of heating rate). The resulting char was cleaned with 6 M HCl and washed with deionised water to neutralise the pH. Finally, the KOH-activated carbon was dried in a vacuum oven at 100 °C for several hours. Hereafter, these samples are denoted HTC-PDS_KOH_x, where x corresponds to the relative mass of KOH (1 or 2). The third activation process (3) was an acid-medium chemical activation (with H_3_PO_4_): it followed the exact same procedure and mass ratios of the KOH activation with the exception of the oxidiser used. The powdered product is denoted HTC-PDS_H_3_PO_4__x, where x corresponds to the relative mass of H_3_PO_4_.

### 2.3. Characterisation

The morphology and texture of the samples were analysed with a FEI Quanta3D scanning electron microscope (SEM) operated at 10 kV. The structure of the carbons was assessed through Raman spectroscopy using a WITec Alpha300RA spectrometer with an excitation wavelength of 488 nm. Elemental carbon, hydrogen, nitrogen, sulfur and oxygen were quantified in wt %, using a Flash2000 (CHNS/O) analyser (ThermoScientific, Waltham, MA, USA). Potassium was quantified through inductively coupled plasma optical emission spectroscopy (ICP-OES) in an Agilent 5110 with an asynchronous dual detector configuration and at wavelengths between 167 and 780 nm. The samples were prepared using a microwave-assisted acid digestion method (oxidisers: HNO_3_ and H_2_O_2_). The porosity of the powders was studied using N_2_ adsorption–desorption isotherms run at −196 °C in a Quantachrome Nova 4200e, Boynton Beach, FL 33426 USA. Prior to the adsorption, the samples were degassed overnight at 150 °C. This ensured the removal of inadvertently adsorbed molecular species. The mesopores specific surface area (S_BET_, where BET stands for Brunauer-Emmet-Teller), the micropores specific surface area (S_DR_, where DR stands for Dubinin-Radushkevich) and the total pore volume (PV) were determined from the N_2_ isotherms and followed the standards of the International Union of Pure and Applied Chemistry (IUPAC). The pore size distribution was obtained by fitting the N_2_ isotherms with models for slit pores, based on the quenched solid density functional theory (QS-DFT) and included in the Quantachrome NovaWin software version 11.03.

### 2.4. CO_2_ Adsorption

The CO_2_ adsorption capacity of the HTC-PDS powders was investigated in a Quantachrome Nova 4200e at 1 bar and 25 °C. Prior to the adsorption runs, the samples were first degassed overnight at 150 °C. The adsorption isotherms shown in this paper were taken after running at least three adsorption-desorption cycles. In between each cycle, the materials were degassed at 150 °C.

The adsorption capacity (*Ca*) was calculated using the formula:(1)$$Ca=\frac{Volume\;of\;gas\;adsorbed\;\left(\frac{cc}{g}\right)}{Molar\;volume\;\left(\frac{cc}{mmol}\right)}$$

For the volume of gas adsorbed, the maximum cm^3^ g^−1^ value registered in the N_2_ isotherms was considered. The *Ca* values are reported in millimoles (mmol) of adsorbed gas per gram of the HTC-PDS powder adsorbent.

## 3. Results and Discussion

In the literature, it is common that char activation is performed via a particular physical or chemical method. However, presenting CO_2_ adsorption studies without contextualizing the carbonaceous synthesis path may lead to less-than-optimal sorbent performances. In fact, it is well-known that the properties of carbon powders differ widely depending on their granular texture, graphitization degree and surface characteristics. All of these can be modified during the activation process and affect the sorption response.

### 3.1. Characterization of the PDS Carbon Powders

Considering the above paragraph, and following the preparation of the five activated biomass-derived carbon powders, their structural and chemical characterization was performed. First, the morphology and texture of the samples were examined. Figure 1 shows representative SEM images of the hydrochar (before activation) and the activated samples, denoting how the different processes acted on the PDS. Initially, the carbonised date seeds showed a compact spherical shape without pores (Figure 1a). These particles had diameters ranging from 3 to 10 μm and were nucleated during the HTC process. Similar carbon microspheres, also produced by HTC, were previously reported using varied biomass sources [29,30,31]. As seen in Figure 1b, the physical activation using CO_2_ had no significant impact on the HTC-PDS particle shape and surface morphology. Conversely, the chemical activation with KOH (Figure 1c,d) caused a visible degradation of the surface as a result of the oxidation reaction. The higher hydroxide concentration (HTC-PDS_KOH_2) was particularly effective in introducing surface roughness but did not compromise the spherical shape of the particles. The analogous acid treatment had an outcome similar to that of the gas process (Figure 1e,f). No visible impact resulted on the particles’ shape and surface morphology.

Apart from the morphology, it is critical to understand how the carbon lattice is structured in these particulate powders. In this respect, Raman spectroscopy is a powerful analytical tool for studying the lattice structure of carbon materials. In Figure 2, the different spectra of the activated powders reflect some common features. First, the two characteristic bands of graphitic carbons were centred at 1338 cm^−^^1^ (D band) and 1585 cm^−^^1^ (G band) [32]. Throughout the samples, there were no meaningful shifts in the wavenumbers of these bands (Appendix A), with the exception of HTC-PDS_KOH_1 (D band = 1329 cm^−^^1^ and G band = 1564 cm^−^^1^). Their intensity ratio (*I_D_*/*I_G_*) and band overlap were used to infer the density of lattice defects. Generally, the ratio increased slightly when a higher concentration of the oxidiser was used (on the KOH and H_3_PO_4_ samples). The band overlap was similar, apart from the sample HTC-PDS_KOH_1, where the separation was more visible, as its bands had reduced widths. Furthermore, both KOH-activated powders had a visible 2D peak at ca. 2658 cm^−1^. The Raman analysis confirmed that the CO_2_ and H_3_PO_4_ activations produced similar carbon materials (in regards to morphology and structure). By contrast, wet impregnation with the hydroxide leads to a better volumetric structural ordering (stacking of graphene layers), particularly for the sample HTC-PDS_KOH_1, where the 2D is quite noticeable. On the whole, the surface roughness and structural order differ in the two HTC-PDS_KOH samples, with the smoother surface HTC-PDS_KOH_1 showing a higher graphitization degree.

Apart from the morphological and structural features, it was important to understand the chemical composition of the activated HTC-PDS. The quantification of C, O, N and H is presented in Table 1 (in wt %), where the C/O ratios are included. The N and H mass fractions were relatively small for all samples, in particular for those exposed to alkaline wet impregnation. Remarkably, it was also the HTC-PDS_KOH powders that had the highest (HTC-PDS_KOH_1) and lowest (HTC-PDS_KOH_2) content of C; the same happened for the O mass fraction but in the inverse proportion. For that reason, the maximum C/O ratio of 7.7 was seen for HTC-PDS_KOH_1, whereas the minimum of this ratio (1.4) was observed in the HTC-PDS_KOH_2 sample. All other samples had mass fractions and C/O ratios that fell between these extremes. Therefore, the elemental analysis reveals that the KOH wet impregnation was the most efficient approach to promote the oxidation of the HTC-PDS carbon lattice, but its yield was highly dependent on the initial concentration of the (alkaline) oxidiser. The same was not observed for the acid treatment, where doubling the amount of H_3_PO_4_ did not change the final C/O ratio.

Overall, the elemental analysis results match well with the SEM and Raman data. The HTC-PDS_KOH_1 sample had the most ordered structure and this is consistent with a chemical profile that has the maximum C/O ratio. Contrastingly, the HTC-PDS_KOH_2 sample—with its spherical particles showing a roughed surface and fairly disordered carbon lattice—was logically the one with the highest oxygen mass fraction and minimal C/O ratio.

### 3.2. Porosity of the PDS Carbon Powders

The assessment of the porosity in activated carbon materials is key to understanding their capability to adsorb gases. In Figure 3, the nitrogen adsorption–desorption isotherms (taken at −196 °C) and the pore size distributions of the HTC-PDS (non-activated) of the various activated powders are shown. Whilst the parent HTC-PDS showed negligible N_2_ sorption, the main textural properties of the activated samples could be calculated and are summarised in Table 2. For comparison, the isotherms of the raw PDS powder are presented in Appendix A.

According to the IUPAC’s nomenclature, all the materials exhibited a type IV adsorption isotherm and a type H4 hysteresis loop in the relative pressure range of 0.4 to 1.0 (Figure 3a). Generally, a type IV isotherm is associated with a well-developed network of mesopores in a powdered material, whilst a type H4 hysteresis loop indicates the presence of slit-shaped mesopores (characteristic of graphitic carbons) [33,34]. It follows from the isotherms and Table 2, that the samples activated with KOH were clearly superior in terms of surface area and pore volume, with the HTC-PDS_KOH_2 sample having the best N_2_ adsorption performance. The analysis of the pore widths revealed that all the activated powders had a wide pore size distribution (PSD), although there was a common dominant interval between 3.5 nm and 5 nm (Figure 3b). Once again, the alkaline oxidation gave the best results. In fact, the HTC-PDS_KOH_2 sample stands out due to a profile that combined micropores (<2 nm) and mesopores that were slightly larger than those of the rest of the samples. An additional noteworthy result is that the CO_2_ activation was the least effective in inducing the porosity in the PDS hydrochar.

It has been observed that increasing the activator-to-hydrochar ratio can lead to notable increments in the surface area and pore volume (textural properties) [35]. In the specific case of HTC-PDS, this is attributed to the enhancement of the surface etching of the spherical carbon particles, which ultimately creates mesopores and a random network of these pores (in addition to microporosity) [36,37]. Here, this same action was observed, as the unique mesoporous structure of the HTC-PDS_KOH_2 sample contributed to its higher specific surface area and bimodal pore size distribution. In addition to porosity considerations, it is also possible that the aforementioned lattice defects and/or an O-rich surface chemistry assisted with physically retaining a higher volume of the analytical N_2_ gas (at −196 °C). Thus, in regards to the surface area and pore volume, the HTC-PDS_KOH_2 sample was clearly the dominant sample, as can be easily seen in the plot of Appendix A.

### 3.3. CO_2_ Adsorption Studies

The CO_2_ adsorption capacities of the activated HTC-PDS materials were measured by the adsorption isotherms at 25 °C under a maximum pressure of 1 bar, as shown in Figure 4. Interestingly, in the lower pressure interval (<0.4), the PDS_CO_2_ activation surpassed all other samples. In this regime, only the most exposed surfaces would be subjected to interaction with the greenhouse gas. As the pressure builds up, the HTC-PDS_KOH_1 started to stand out, eventually reaching a maximum value of 5.44 mmol g^−1^ at 1 bar (Figure 4 and Appendix A). Conversely, the HTC-PDS_H_3_PO_4__2 sample provided a minimum of 4.00 mmol g^−1^. As for the other samples, they had similar uptake values in the high-pressure range (ca. 4.50 mmol g^−1^). Hence, within the set of samples investigated, there was a differentiation in the CO_2_ uptake that is not only correlated with the type of activation but also with the concentration of the oxidiser.

### 3.4. Discussion

The CO_2_ uptake of HTC-PDS was studied as a function of the activation approach and concentration of oxidiser. Whilst the morphology of the activated carbon particles was equal (all had a spherical shape), the chemical composition differed (viz. the C/O ratios in Table 1). Alongside, there were notable variations in the specific surface area, microporosity and total pore volume. Previously, some of us looked into the total adsorption capacity of CO_2_ using nanocarbons—specifically, platelets of oxidised graphite [34]. Then, it was concluded that the uptake was not only a function of the specific surface area, but also of the particles’ morphology and surface chemistry.

As described above, the best performance was achieved with the alkaline wet activation. The mechanism of KOH activation was explained by Wang and Kaskel [38], and can be summarised in three steps: (1) an etching (oxidation) of the atomic carbon lattice with the formation of vacancy clusters via redox reactions; (2) a structural disruption (porous structure development) due to high-pressure pockets originating from a build-up of H_2_O and CO_2_ vapors (these are redox reaction products); (3) if graphitic crystallites are present, an expansion of the basal planes via the intercalation of potassium cations and formation of slit-shaped extrinsic micropores.

In the present case, whilst the observed differences of surface area and pore volume were important, they do not explain the superiority of the HTC-PDS_KOH_1 sample. In regards to the textural properties, this sample was worse than the analogous HTC-PDS_KOH_2 (Appendix A). Two differences stand out between them: the C/O ratio and the “graphitization” degree. In both cases, the HTC-PDS_KOH_1 sample was clearly superior. The lower concentration of oxygen in this activated carbon powder is understandable, given that half of the KOH oxidiser was used for this sample. For this reason, the oxidiser was fully consumed and the extent of etching (as per step 1 above) was lower, leading to less pore development and allowing the carbon lattice to restructure more efficiently (i.e., carbonization at 900 °C and under a N_2_ stream). Ultimately, this resulted in the formation of graphitic crystallites in higher amount/size, as exhibited by the Raman spectrum of the HTC-PDS_KOH_1 sample. An immediate consequence of such superior crystalline order was the maximal production of the extrinsic pores mentioned in step 3. With more and better structured graphene layers, the probability of the intercalation of potassium increases. As it is unlikely that the alkali cation would be reduced, its intercalation would lead to a change in the local density of states (LDOS) of the graphene layers. In particular, an electronic depletion of the surface graphene would result in pockets of charge deficit. If so, it is logical that one of the oxygens in the linear CO_2_ molecule would anchor preferably to such a site, most likely one with a higher adsorption energy. Consequently, the CO_2_ molecule quadrupole moment would be altered, facilitating any further adsorption and contributing to a larger gas uptake. By contrast, the non-structured porous carbon in the HTC-PDS_KOH_2 sample did not permit such LDOS changes and hence its adsorption capacity was lower. Thus, despite its worse textural properties (surface area and pore volume), the HTC-PDS_KOH_1 sample was superior in the adsorption of CO_2_ due to the complementary effects of pore development and lattice restructuring. These improved adsorbate–adsorbent electronic interactions are more visible at higher pressures due to the sequential stacking of adsorbed gas on the acceptor-type graphene surfaces. To support this hypothesis, we measured the concentration of potassium in the parent HTC-PDS and the two HTC-PDS_KOH samples. As seen in Table 1, the presence of the alkali metal was considerably higher in the activated powders, even after the thorough post-activation washing steps (with HCl and water). Despite the HTC-PDS_KOH_2 having more KOH in its synthesis mass ratio (hydrochar: KOH, 1:2), it was the HTC-PDS_KOH_1 sample (with 1:1) that showed the highest content of K. This suggests the inclusion of potassium in the microspheres’ surface crystallites, with the additional potassium identified in the HTC-PDS_KOH_1 sample being located in the interstitial spaces (thus resulting in the formation of extrinsic lattice pores) [39].

Given the above, it is useful to compare our results with the literature of biomass-derived activated carbon adsorbents. Table 3 shows the performance of several of these materials (reported for room temperature and 1 bar). Whilst our HTC-PDS_KOH_1 powder had good textural properties, it is not truly outstanding. Still, its CO_2_ uptake was superior. It is difficult to pinpoint a reason for this. However, reading through the literature, the study of the lattice ordering is a critical point often overlooked. When it is discussed, it is not uncommon that the Raman spectra are dominated by the D-peak and there is an absence of the 2D peak [40]. Furthermore, for those studies using KOH (or other chemical agents), it is rare to see the study of elemental composition. That means that step 3 of the above mechanism is generally ignored or not discussed with proper supporting evidence. In this context, and given the information available, the crystallites with intercalated potassium cations could indeed contribute to the adsorption of CO_2_ through two mechanisms: (1) the co-intercalation of CO_2_ molecules and (2) a surface process akin to the formation of an electrostatic double-layer such as that seen in carbon-based supercapacitors [41].

## 4. Conclusions

Porous carbon materials from palm date seeds have been synthesised via HTC combined with physical or chemical activation. It is clear that, besides the activation route, selecting the appropriate activator ratio was critical to controlling the yield of the CO_2_ uptake in these samples. Overall, as a result of well-balanced and synergetic effects of chemical oxidation, pore development, lattice restructuring and potassium doping, the HTC-PDS_KOH_1 powder was optimised to adsorb CO_2_ at room temperature (5.44 mmol g^−1^, 25 °C, 1 bar). Finally, it is promising that the best performance was achieved for the sample activated with the smallest amount of the chemical agent, since this reinforces the sustainability credentials of the entire process.

## Figures and Tables

**Figure 1 ijerph-18-12142-f001:**
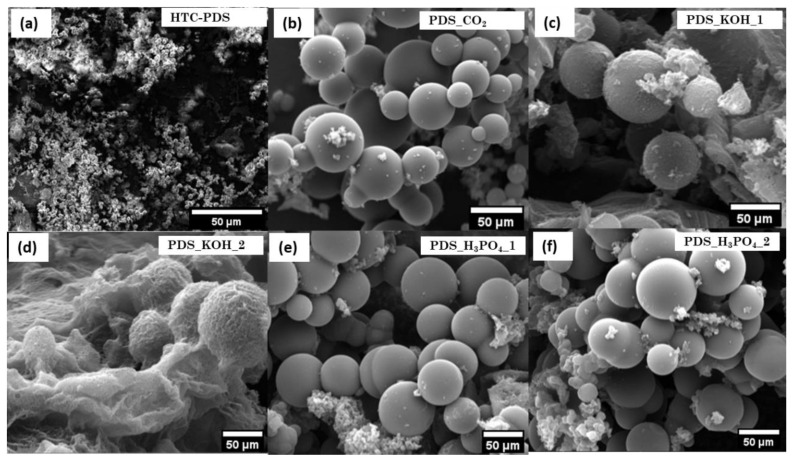
SEM images of the (**a**) HTC-PDS and (**b**–**f**) the corresponding activated materials.

**Figure 2 ijerph-18-12142-f002:**
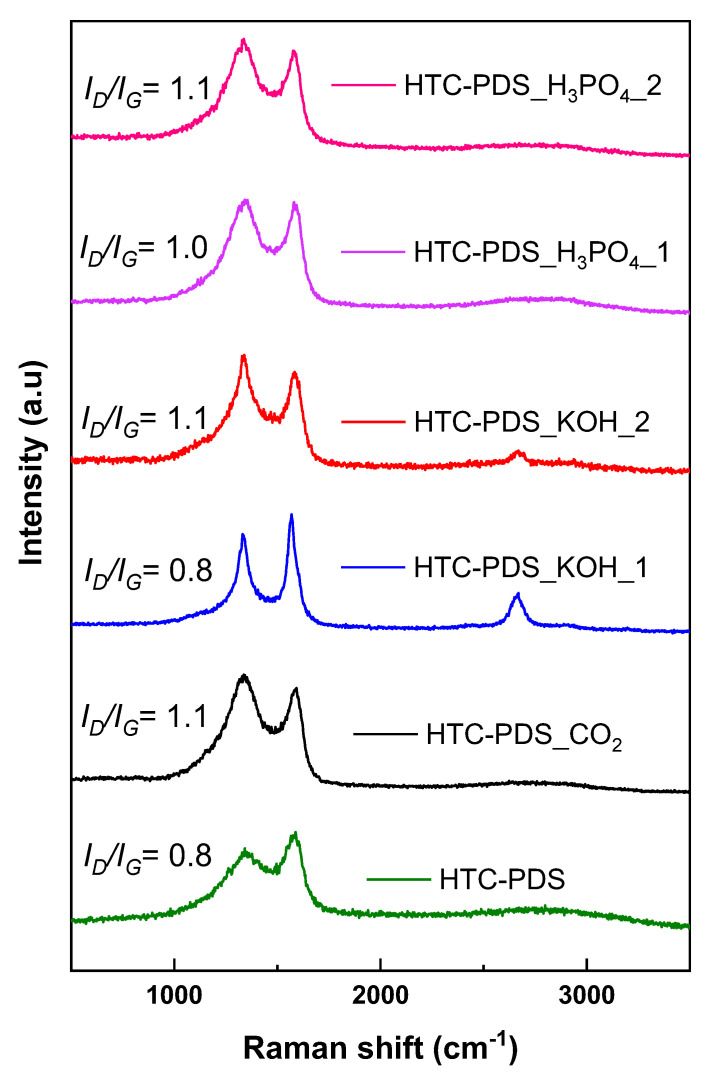
Raman spectra for the HTC-PDS and the respective activated carbon materials.

**Figure 3 ijerph-18-12142-f003:**
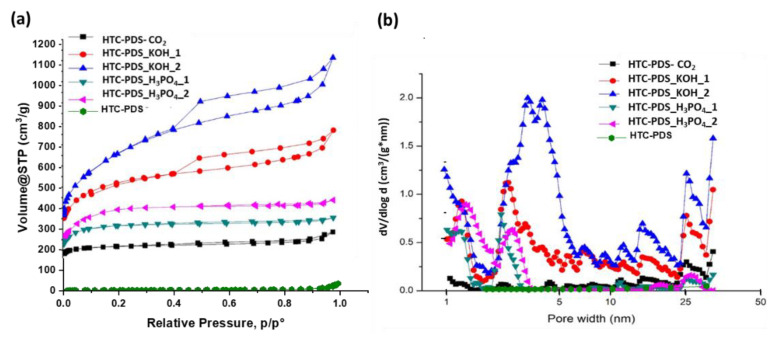
(**a**) N_2_ sorption isotherms and (**b**) the pore size distributions of the HTC-PDS and respective activated materials. The measurements were performed at −196 °C with N_2_.

**Figure 4 ijerph-18-12142-f004:**
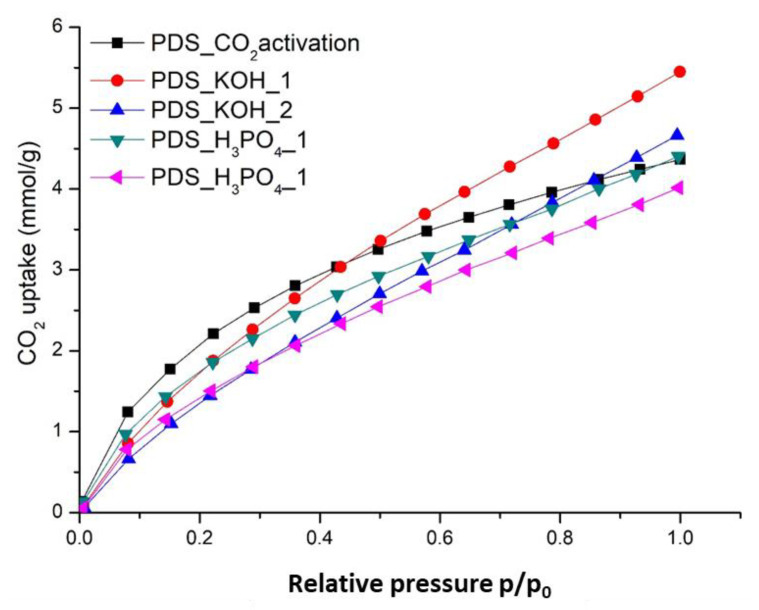
Pure component CO_2_ adsorption isotherms, measured at 25 °C and up to 1 bar, of the activated HTC-PDS powders.

**Table 1 ijerph-18-12142-t001:** Elemental analysis of the parent HTC-PDS and the respective activated materials. n.a. = not available.

Sample	Elemental Composition (wt %)				C/O Ratio	K (ppm)
	C	N	H	O		
HTC-PDS	67.6	1.4	5.1	25.9	2.6	586
HTC-PDS_CO_2_ activation	81.1	1.7	1.4	15.8	5.1	n.a.
HTC-PDS_KOH_1	87.8	0.6	0.3	11.3	7.7	8840
HTC-PDS_KOH_2	57.4	0.4	0.5	41.7	1.4	8415
HTC-PDS_H_3_PO_4__1	69.4	1.4	2.0	27.2	2.5	n.a.
HTC-PDS_H_3_PO_4__2	70.2	1.3	2.0	26.5	2.6	n.a.

**Table 2 ijerph-18-12142-t002:** Textural properties of the HTC-PDS-activated materials. *μV = micropore volume.

Sample	SBET (m^2^ g^−1^)	SDR (m^2^ g^−1^)	SDFT (m^2^ g^−1^)	*μV (cm^3^ g^−1^)	PV (cm^3^ g^−1^)
HTC-PDS_CO_2_	858	949	910	0.34	0.39
HTC-PDS_KOH_1	1906	2189	1867	0.78	1.06
HTC-PDS_KOH_2	2335	2552	2122	0.90	1.54
HTC-PDS_H_3_PO_4__1	1218	1403	1251	0.50	0.50
HTC-PDS_H_3_PO_4__2	1439	1674	1086	0.60	0.60

**Table 3 ijerph-18-12142-t003:** The adsorbent performance of some activated porous carbon materials, generally derived from biomass, at 25 °C and maximum CO_2_ pressure of 1 bar. n.a. = not available. * MIP = molecularly imprinted polymers.

Precursor	Activation Method	BET Surface Area (m^2^ g^−1^)	Pore Volume (cm^3^ g^−1^)	Structure (*I_D_*/*I_G_*)	CO_2_ Adsorption (mmol g^−1^)	Ref.
Commercial activated carbon	-	698	0.21	n.a.	2.18	[42]
Palm date seeds (UAE)	Physical activation (under CO_2_)	798	0.28	n.a.	3.20	[42]
Olive stones/almond shells	Physical activation (under CO_2_)	1113	0.51	n.a.	1.02	[43]
CO_2_–MIP *	Chemical activation (KOH)	-	-	n.a.	1.71	[44]
Camphor leaves	Chemical activation (KOH)	1633	0.98	n.a.	0.80	[14]
Oil-based pitch	Chemical activation (KOH)	1720	0.98	n.a.	1.90	[45]
Activated biocarbon	Chemical activation (KOH)	1968	1.14	0.6	1.67	[40]
Palm date seeds (KSA)	Chemical activation (KOH)	1906	1.06	1.1	5.44	This work

## Data Availability

Not applicable.

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
