# Peer review of "Activated Carbon from Palm Date Seeds for CO2 Capture"

_ijerph, 2021, doi:10.3390/ijerph182212142_

Round 1

Reviewer 1 Report

This study describes the production of activated carbon from palm date seeds by means of hydrothermal carbonization and physical or chemical activation. The area of application is CO2 capture. Palm dates are an interesting source of biomass with increasing use in research. The manuscript is very well written and the results demonstrate the significantly improved properties of these materials compared to the literature.

Nevertheless, some points should be revised or critically examined:

  1. Please check in Figure 1 if the scales are correct. Images b) to f) seem to show a higher magnification than a). It is not clear from the text how activation with CO2/KOH/H3PO4 can explain the size increase of the particles compared to the HTC-PDS powder.
  2. The control/reference "HTC-PDS" (without activation) should be included in the manuscript for easier comparison and not in the SI. Please add the data for HTC-PDS as reference in Table 1, Figure 3, Table 2, and Figure 5, as well as in the text.
  3. Figure 4 is missing in the manuscript. Figure 5 is mentioned first in section 3.3. In the Discussion, Figure 4 is mentioned, which does not exist.
  4. A central role in the discusion is the incorporation of potassium, but this has not been proven (e.g. elemental analysis). This is especially important with regard to the washing process using HCl and water. In the second paragraph of the discussion, the mechanism of KOH activation is described. Please check the correct naming of potassium vs potassium cations (the latter should not be a redox product after all).
  5. In general, it seems that single measurements were perfomed. The lack of confidence intervals/errors, e.g. in Figure 5 or Table 1, makes a statistical sound evaluation of the results difficult.

I hope this is helpful to further improve the manuscript.

Author Response

Response to Reviewer 1 Comments

Point 1: Please check in Figure 1 if the scales are correct. Images b) to f) seem to show a higher magnification than a). It is not clear from the text how activation with CO2/KOH/H3PO4 can explain the size increase of the particles compared to the HTC-PDS powder.

Response 1: We thank the referee for the comments to the manuscript. Note that the SEM images b) to f) were collected at a higher magnification than a). To make this more visible, we updated the scale bars in the Figure. We have measured the activated spherical particles and the parent HTC-PDS microspheres, and they show similar average size.

Point 2: The control/reference "HTC-PDS" (without activation) should be included in the manuscript for easier comparison and not in the SI. Please add the data for HTC-PDS as reference in Table 1, Figure 3, Table 2, and Figure 5, as well as in the text.

Response 2: Along with Figure 3, Table 1 has been updated with the HTC-PDS data. Regarding the HTC-PDS data for Table 2 and Figure 5 (now, Figure 4), giving the fact that the control material exhibited very low surface area (i.e. microporosity) as seen in Figure 3, we did not proceed to measure the CO2 uptake for this sample.

Point 3: Figure 4 is missing in the manuscript. Figure 5 is mentioned first in section 3.3. In the Discussion, Figure 4 is mentioned, which does not exist.

Response 3: Thank you for highlighting this. There was a mislabelling and Figure 4 should be Figure S2 (now corrected).

Point 4: A central role in the discusion is the incorporation of potassium, but this has not been proven (e.g. elemental analysis). This is especially important with regard to the washing process using HCl and water. In the second paragraph of the discussion, the mechanism of KOH activation is described. Please check the correct naming of potassium vs potassium cations (the latter should not be a redox product after all).

Response 4: These are very valid comments, thank you. On the first, we have now carried out the elemental analysis for potassium (with ICP-OES). The parent HTC-PDS shows 586 ppm, while the activated HTC-PDS_KOH_1 reads 8840 ppm. Interestingly, this is a higher concentration than the one read for the analogous HTC-PDS_KOH_2. Despite its higher C/O ratio, the sample of 1:2 ratio as a lower presence of K. We interpret this as circumstantial evidence of intercalation, also supported by previous literature that refers the intercalation of potassium in carbon microspheres. However, we cannot assert directly this statement. On the second point, effectively, given the minute probability that the potassium cation (from the KOH salt) will be reduced to the elemental form, it is more correct to specify that the species is the cation. The sentence has now been changed to: “(…) 3) if graphitic crystallites are present, expansion of the basal planes via intercalation of potassium cations and formation of slit-shaped extrinsic micropores.”

Point 5: In general, it seems that single measurements were perfomed. The lack of confidence intervals/errors, e.g. in Figure 5 or Table 1, makes a statistical sound evaluation of the results difficult.

Response 5: We agree with this comment and understand the need to inform on this. In fact, we have carried out, at least, three measurements for each sample. This was to assert stability and reversibility of uptake. However, as the scope of the study was to identify the best activation route, we kept the data only for the last adsorption isotherm, the one shown. After the present activation comparison study, we intend to explore further our best sample and will explore things such as reversibility and selectivity at different temperatures.

Reviewer 2 Report

Comments:

  1. After the CO2 capture by the activated carbon of palm date seeds, how to treatment the used activated carbon of palm date seeds ? Maybe the CO2 release to ambient again. Please illustrate detail about the application of adsorpted-CO2.
  2. Please discuss the effect of CO2 adsorption at different environment factors (ex: ambient temperature). Then provide the optima operation conditions for CO2 capture by the activated carbon of palm date seeds for real-word application.
  3. Please estimate the cost ($/g-CO2) of making the activated carbon of palm date seeds for CO2 capture.

Author Response

Response to Reviewer 2 Comments

Point 1: After the CO2 capture by the activated carbon of palm date seeds, how to treatment the used activated carbon of palm date seeds ? Maybe the CO2 release to ambient again. Please illustrate detail about the application of adsorpted-CO2.

Response 1: We thank the referee for the comments made to the manuscript. On this particular point, one of the measurements we carried out was at ambient temperature. to study the behaviour of the activated material at room temperature. From figure 4, it is clear that the samples show relatively high CO2 uptake at room temperature. Note that the adsorption isotherms shown were taken after running, at least, three adsorption-desorption cycles. In between each cycle, the materials were degassed at 150 °C. While this study explored the process of activation, the follow-up work will focus on the properties of the optimised material.

Point 2: Please discuss the effect of CO2 adsorption at different environment factors (ex: ambient temperature). Then provide the optima operation conditions for CO2 capture by the activated carbon of palm date seeds for real-word application.

Response 2: We interpret this comment as a request to consider variables such as cycling and selectivity, at different temperatures and gas environments. It is effectively something that we plan to do, i.e. delving further into the optimised sample properties. Note that the current study served to identify the best activation route, and activator ratio, for the HTC-PDS chars.

Point 3: Please estimate the cost ($/g-CO2) of making the activated carbon of palm date seeds for CO2 capture.

Response 3: Considering the type of biomass used, i.e. palm date seeds, these were available at no cost. The pits are generally discarded as waste by farmers and various processing units of the date fruit. At laboratory scale, the synthesis process yield was approximately 7%. In this way, we can proceed with the following example:

  • The biomass is free
  • Experimental measurements showed that from 1.5 g of mix (hydrochar and KOH activator in equal mass ratio), one can obtain 0.1 g of activated carbon (CO2 adsorbent), meaning a 6.66% yield
  • From the above, in order to obtain 1 g of activated carbon (AC), 15 g of the mix would be necessary, from which half would be the KOH (7.5 g); according to Merck, the price for 1 kg of KOH is 40.30 euros; therefore, 7.5 g will cost 0.302 euros. Therefore, it results that the price of KOH for 1 g AC (for CO2 adsorption) is $0.35
  • The samples were washed with approximately 100 mL HCl (6 M); here, we will assume that approximately 1 L of HCl (6M) will be necessary to wash 1 g of AC. Commercially, the cost of 1 L of HCl (6M) is approximately $48
  • For utilities wear and other indirect costs (e.g. water, electricity), this will be highly variable, so it is difficult to judge
  • Overall, we estimate that the cost for producing 1g of the activated HTC-PDS will be approximately $50 (+ indirect costs)

Round 2

Reviewer 1 Report

The authors revised the manuscript and gave sufficient consideration to the comments. 

Reviewer 2 Report

No comments.